

# Differential response to stress in *Ostrea lurida* as measured by gene expression

J. Emerson Heare, Samuel J. White, Brent Vadopalas and Steven B. Roberts

School of Aquatic and Fishery Sciences, University of Washington, Seattle, WA, United States of America

## ABSTRACT

Olympia oysters are the only oyster native to the west coast of North America. The population within Puget Sound, WA has been decreasing significantly since the early 1900's. Current restoration efforts are focused on supplementing local populations with hatchery bred oysters. A recent study by *Heare et al. (2017)* has shown differences in stress response in oysters from different locations in Puget Sound however, nothing is known about the underlying mechanisms associated with these observed differences. In this study, expression of genes associated with growth, immune function, and gene regulatory activity in oysters from Oyster Bay, Dabob Bay, and Fidalgo Bay were characterized following temperature and mechanical stress. We found that heat stress and mechanical stress significantly changed expression in molecular regulatory activity and immune response, respectively. We also found that oysters from Oyster Bay had the most dramatic response to stress at the gene expression level. These data provide important baseline information on the physiological response of *Ostrea lurida* to stress and provide clues to underlying performance differences in the three populations examined.

## INTRODUCTION

Olympia oysters, *Ostrea lurida*, are the only native oyster species on the west coast of North America. The species inhabits bays and estuaries within Puget Sound, WA. *Ostrea lurida* is typically smaller than the introduced Pacific oyster, *Crassostrea gigas*, with adults attaining an average size between 40 and 60 mm (*Hopkins, 1936*; *Baker, 1995*). As protandric hermaphrodites, Olympia oysters usually spawn as both male and female within the first year (*Coe, 1932*; *Hopkins, 1936*; *Baker, 1995*). Unlike *C. gigas*, *O. lurida* does not release its eggs into the water column. Instead females collect planktonic sperm balls and larvae are brooded for approximately two weeks before being released into the water column. The adults are sessile and are typically only moved via predator interactions or wave action. Colonizing lower intertidal habitats, *O. lurida* typically can be found in the inner portions of bays or estuaries where dynamic conditions can shape the phenotypes of local populations (*Baker, 1995*; *White, Ruesink & Trimble, 2009*).

Loss of habitat due to invasive species, overharvest, and pollution have greatly reduced the native Olympia oyster population. Although restoration efforts are underway, basic research is needed to understand how this species interacts with its environment and responds to stress. Freshwater influx, tidal exchange, food availability, shifts in water temperature,

Corresponding author
Steven B. Roberts, sr320@uw.edu

and physical stresses from water flow and predation are examples of a myriad of stressors which affect long term survival of *O. lurida* populations (*Hopkins, 1936*; *Baker, 1995*).

Thermal stress has been widely studied in mollusks, especially bivalves. *Ostrea lurida* has a temperature tolerance range between 5 °C and 39 °C (*Hopkins, 1936*; *Brown et al., 2004*). It is suspected that mass summer mortalities of *C. gigas* may be linked to the effects of heat stress during spawning events (*Li et al., 2007b*). The California mussel, *Mytilus californianus*, has been found to divert resources to physiological defense during thermal stress events (*Petes, Menge & Harris, 2008*; *Fitzgerald-Dehoog, Browning & Allen, 2012*). Expression of homeostasis-related genes, such as HSP70, glutamine synthetase, and citrate synthase in *C. gigas* has been shown to fluctuate under prolonged heat stress at 25 °C for 24 days (*Meistertzheim et al., 2007*). Temperature stress has been shown to induce a variety of up and down regulation of genes to maintain homeostasis (*Tomanek, 2010*). In oysters, there has been a significant amount of work examining the change in heat shock protein (HSP) family gene expression. Seasonal variation of HSPs and heat shock cognates (HSCs) levels have been characterized in response to ambient temperatures for *C. gigas* (*Hamdoun, Cheney & Cherr, 2003*; *Farcy et al., 2009*). Additionally, induction of HSP70 and HSP69 in *Ostrea edulis* at temperatures greater than 38 °C have been reported (*Piano et al., 2005*).

The response of bivalves to mechanical stress has also received considerable attention. One reason for this is that researchers have shown mechanical stress elicits a classical stress response, providing a simple method to allow for investigation of fundamental physiological stress responses. Additionally, most oyster restoration and aquaculture practices do involve handling and movement which would be a form of mechanical stress. Mechanical stress in oysters has been shown to increase catecholamines present in hemolymph (*Qu et al., 2009*; *Lacoste et al., 2001c*). Upon mechanical stress, researchers have found increases in adrenocorticotropic hormone (ACTH), a hormone that induces production of noradrenaline and dopamine (*Lacoste et al., 2001a*; *Lacoste et al., 2001b*; *Lacoste et al., 2001c*). Mechanical stress has also been shown to activate inflammation factors that are also observed during bacterial challenges (*Lacoste et al., 2001c*; *Lacoste et al., 2001d*; *Aladaileh, Nair & Raftos, 2008*; *Roberts, Sunila & Wikfors, 2012*). Studies in Pearl oysters (*Imbricata pinctada*) have found significant decreases in phagocytosis and phenoloxidase activity due to mechanical stress (*Kuchel, Raftos & Nair, 2010*).

Here we set out to examine the effects of temperature and mechanical stress on *Ostrea lurida*, by comparing differences in gene expression among three local populations (*Heare et al., 2017*). Each of the three populations comes from distinct bays within Puget Sound, WA: Fidalgo Bay, Dabob Bay, and Oyster Bay. Fidalgo Bay, the furthest northern population (48°28′31.1″N 122°34′48.6″W), is directly fed from the Salish Sea and the Strait of Juan de Fuca, and has the coldest average year-round temperatures of the three locations. Typically, this population does not experience strong fluctuations in temperatures due to the fact that it resides in the lower part of the intertidal area and is submerged for most of the time. Olympia oysters from Fidalgo Bay experience significant growth when placed in warmer habitats, but otherwise lack other observable phenotypes (*Heare et al., 2017*). Dabob Bay (47°49′27.4″N 122°48′37.9″W) is a large bay at the northern most portion of Hood Canal with the population of Olympia oysters residing near the innermost portions of the bay

(e.g., Tarboo Creek). This area experiences extreme temperature fluctuations throughout the year and this population of *O.lurida* is often partially, or completely, exposed during low tide events. During tidal changes, temperatures can be as high as 29 °C during summer or as low as −3 °C during winter (*Heare et al., 2017*). Oysters from Dabob Bay have been shown to experience high survival when faced with temperature challenges, possibly due to adaptive structure of the local population (*Heare et al., 2017*). Oyster Bay (47°06′21.2″N 123°04′32.8″W) is the southernmost bay which sustains a healthy population of *O. lurida*. The conditions here are, on average, the warmest of the three locations throughout the year. The bay has extensive food resources and oysters appear to allocate more energy resources into reproductive activity compared to the other populations, based on our prior field studies (*Heare et al., 2017*).

For long-term restoration of *O. lurida* populations in Puget Sound, understanding the phenotypic plasticity of individual populations will help determine proper supplementation procedures for existing and historic habitats. To this end, and to attempt reveal relationship of gene expression response with stress exposure, we investigated differences between these populations in their responses to mechanical and temperature stresses, based on mRNA expression of select target genes as measured by quantitative PCR (qPCR). A suite of genes was selected based on their predicted functions related to gene regulation, immune response, and growth. Given the field performance of these populations, we hypothesized we would see differences in response that could be indicative of underlying genetic population differences. A specific hypothesis is that oysters from Dabob Bay will demonstrate a more pronounced response to stress via changes in gene expression.

## MATERIALS AND METHODS

### Experimental design

Adult, hatchery produced oysters from three wild source populations (Dabob Bay, Fidalgo Bay, and Oyster Bay grown for 19 months at Clam Bay) in WA were used for this experiment. All oysters were held at 8 °C for two weeks at the University of Washington prior to the experiment. Oysters from each population ($n = 8$ per population) were subjected to acute temperature stress (submerged in 500 mL 38 °C sea water for 1 h), mechanical stress (120 g × 5 min; Sorvall T21, ST-H750 rotor) or served as controls (maintained at 8 °C). After the stress treatments, oysters were returned to 8 °C seawater and sampled at 1 h post stress ($n = 72$). Ctenidia tissue was resected from each individual and stored separately in 500 μL RNAzol RT (Molecular Research Center, Inc.), frozen on dry ice. All samples were stored at −80 °C for later analysis.

### RNA isolation

RNA was isolated using RNAzol RT (Molecular Research Center, Inc., Cincinnati, OH, USA) according to the manufacturer's protocol for total RNA isolation. Briefly, ctenidia tissue was homogenized in RNAzol RT, volume was brought up to 1 mL with RNAzol RT, vortexed vigorously for 15 s, and incubated at room temperature (RT) for 10 min. 400 μL of 0.1% DEPC-$H_2O$ was added to the homogenized ctenidia tissue, vortexed for 15 s, and incubated at RT for 15 min. The samples were centrifuged for 15 min, 16,000 g, at RT.

After centrifugation, 750 μL of the supernatant was transferred to a clean tube, an equal volume of isopropanol added, vortexed for 10 s, and incubated at RT for 15 min. The samples were centrifuged at 12,000 g for 10 min at RT. The supernatant was discarded and the pellets were washed with 500 μL of 75% ethanol (made with 0.1% DEPC-$H_2O$) and centrifuged at 4,000 g for 3 min at room temperature. This wash step was then repeated. Ethanol was removed and pellets were resuspended in 100 μL of 0.1% DEPC-$H_2O$. Samples were quantified using a NanoDrop1000 (ThermoFisher, Waltham, MA, USA) and stored at −80 °C.

## DNase treatment and reverse transcription

Total RNA was treated with DNase to remove residual genomic DNA (gDNA) using the Turbo DNA-free Kit (Ambion/Life Technologies, Carlsbad, CA, USA). The manufacturer's rigorous protocol was followed. Briefly, 1.5 μg of total RNA was treated in 0.5 mL tubes in a reaction volume of 50 μL. The samples were incubated with 1 μL of DNase for 30 min at 37 °C. An additional 1 μL of DNase was added to each sample and incubated at 37 °C for an additional 30 min. The DNase was inactivated with 0.2 volumes of the inactivation reagent according to the manufacturer's protocol. Samples were quantified using a NanoDrop1000 (ThermoFisher, Waltham, MA, USA). Treated RNA was verified to be free of gDNA via qPCR using actin primers (see Primer Design section below) known to amplify gDNA.

Reverse transcription was performed using M-MLV Reverse Transcriptase (Promega) with oligo dT primers (Promega, Madison, WI, USA), using 250 ng of DNased RNA. The RNA was combined with primers (0.25 μg) in a volume of 74.75 uL, incubated at 70 °C for 5 min in a thermal cycler without a heated lid (PTC-200; MJ Research), and immediately placed on ice. A master mix of 5× Reverse Transcriptase Buffer (1× final concentration; Promega), 10 mM each of dNTPs (0.5 mM final concentration of each dNTP; Promega, Madison, WI, USA), and M-MLV Reverse Transcriptase (50 U/reaction) was made and 25.25 μL of the mix was added to each sample (final reaction volume 100 μL). Samples were incubated at 42 °C for 1 h, followed by 95 °C for 3 min in a thermal cycler without a heated lid (PTC-200; MJ Research, Waltham, MA, USA), and then stored at −20 °C.

## Quantitative PCR
### Primer design

Primers for qPCR analysis were developed from an *O. lurida* transcriptome (version 3) which can find in the repository associated with this manuscript (*Roberts, 2017*). This transcriptome was annotated using SwissProt and Gene Ontology Databases. Specifically, gene function annotations were based on the protein in the UniProt/SwissProt database that had highest homology with the Olympia oyster sequence (i.e., top Blastp hit). Gene targets were selected based on annotations related to gene regulation, immune response, and growth (Table 1). Corresponding contigs were then selected from the transcriptome using the seqinR package (*Charif & Lobry, 2007*). NCBI Primer Blast was used to develop primers for qPCR using the following parameters: amplicon size 100–400 bp, GC content 55–60%, melt temperatures ∼60 °C and within 0.5 °C of each other, self and 3′ complementarity was limited to 4.00 or less with smallest values being selected, primer sequence 19–21 bp in length.

**Table 1   Table of genes of interest.** The table lists the source transcriptome contigs (annotated by BLASTx against the Uniprot database), as well as the biological categorization, the Uniprot Accession, Uniprot Entry Name, Uniprot Annotation, a brief description of the proteins' functions, and the BLASTx *e*-values.

| Transcriptome contig name | Biological category | Uniprot accession | Uniprot entry name | Uniprot annotation | Function | Gene abbr | BLASTX evalue |
|---|---|---|---|---|---|---|---|
| comp7220_c0_seq2 | Immune response, Gene regulation | Q6DC04 | CARM1_DANRE | Histone-arginine methyltransferase | Transfers methyl groups to Histone 3 for chromatin remodeling | CARM1 | 0 |
| comp23747_c0_seq1 | Immune response | Q9DD78 | TLR21_CHICK | Toll-like receptor 2 type 1 | Assists with recognition of foreign pathogens and endogenous materials for consumptions by phagocytes in early stages of inflammation | TLR | 8.00E−29 |
| comp25000_c0_seq1 | Immune response, Gene regulation | P08991 | H2AV_STRPU | Histone H2A.V | One of 5 main Histone Proteins involved in the structure of chromatin and the open reading frame of DNA | H2AV | 5.00E−64 |
| comp24065_c0_seq1 | Immune response | O75594 | PGRP1_HUMAN | Peptidoglycan recognition protein 1 | Assists with recognition of bacteria in an immune response | PGRP | 2.00E−42 |
| comp44273_c0_seq2 | Immune response | Q8MWP4 | Q8MWP4_OSTED | Heat Shock Protein 70 kDa | Molecular chaperone and protein preservation in heat response | HSP70 | 0 |
| comp7183_c0_seq1 | Growth | P12643 | BMP2_HUMAN | Bone morphogenetic protein 2 | Directs calcification in shell creation | BMP2 | 2.00E−93 |
| comp10127_c0_seq1 | Gene regulation | P62994 | GRB2_RAT | Growth factor receptor-bound protein 2 | Assists in signal transduction/cell communication | GRB2 | 1.00E−83 |

**Table 1** (*continued*)

| Transcriptome contig name | Biological category | Uniprot accession | Uniprot entry name | Uniprot annotation | Function | Gene abbr | BLASTX evalue |
|---|---|---|---|---|---|---|---|
| comp6939_c0_seq1 | Immune response | P32240 | PE2R4_MOUSE | Prostaglandin E2 receptor EP4 subtype | Receptor for Prostaglandin E2 which suppresses inflammation due to injury | PGEEP4 | 1.00E−50 |
| comp25313_c0_seq1 | Immune response | Q60803 | TRAF3_MOUSE | Tumor Necrosis Factor receptor-associated factor 3 | Related to immune response specifically cell death initiation | TRAF3 | 3.00E−145 |
| comp30443_c0_seq2 | Growth | Q8TA69 | Q8TA69_CRAGI | Actin | Cytoskeletal formation. Used as a normalizing gene for qPCR analysis. | Actin | 0 |

**Table 2  Table of qPCR Primers for genes of interest.** Includes the Uniprot Entry Name, the Gene Abbreviation used throughout this manuscript, and the forward (FWD) and reverse (REV) primer sequences. Full sequences utilized for primer creation are available (*Heare & Roberts, 2015*).

| Gene abbreviation | FWD | REV |
|---|---|---|
| CARM1 | TGGTTATCAACAGCCCCGAC | GTTGTTGACCCCAGGAGGAG |
| TLR | ACAAAGATTCCACCCGGCAA | ACACCAACGACAGGAAGTGG |
| H2AV | TGCTTTCTGTGTGCCCTTCT | TATCACACCCCGTCACTTGC |
| PGRP | GAGACTTCACCTCGCACCAA | AACTGGTTTGCCCGACATCA |
| HSP70 | TTGTCGCCATTTTCCTCGCT | GTTCCGATTTGTTCCGTGCC |
| BMP2 | TGAAGGAACGACCAAAGCCA | TCCGGTTGAAGAACCTCGTG |
| GRB2 | AACTTTGTCCACCCAGACGG | CCAGTTGCAGTCCACTTCCT |
| PGEEP4 | ACAGCGACGGACGATTTTCT | ATGGCAGACGTTACCCAACA |
| TRAF3 | AGCAGGGCATCAAACTCTCC | ACAAGTCGCACTGGCTACAA |
| Actin | GACCAGCCAAATCCAGACGA | CGGTCGTACCACTGGTATCG |

Primer binding sites were assessed for the presence of single nucleotide polymorphisms (SNPs) via Sanger sequencing. The majority of primer binding sites did not contain any SNPs. Those that did, had only a single SNP and did not appear to impact qPCR data, as there were no noticeable difference in qPCR efficiencies in individuals having a SNP within a primer binding site for a given target.

The list of primers can be viewed in Table 2.

## Quantitative PCR

Quantitative PCR reactions were carried out using Ssofast Evagreen Supermix (BioRad, Hercules, CA, USA). Forward and reverse primers (Integrated DNA Technologies, Coralville, IA, USA) were used at a final concentration of 0.25 μM each. Sample cDNA was diluted (1:9) with molecular-grade water. Nine microliters of diluted cDNA was used as template. Reaction volumes were 20 μL and were run in low-profile, non-skirted, white qPCR plates (USA Scientific) with optically clear lids (USA Scientific, Ocala, FL, USA) in a BioRad CFX Real Time Thermocycler (BioRad, Hercules, CA, USA) and DNA Engine Opticon 2 System (BioRad, Hercules, CA, USA). Cycling conditions were: one cycle of 95 °C for 10 min; 40 cycles of 95 °C for 30 s, 60 °C for 1 min, 72 °C for 30 s. Two qPCR replicates were run for each sample, for each primer set.

## Statistical analysis

To calculate relative expression levels for each gene, cycle quantity ($Cq$) or cycle threshold ($Ct$) values were calculated using BioRad CFX Manager 3.1 (version 3.1.1517.0823, Windows 8.1) and Opticon Manager 3 (Windows 8.1), respectively. This was accomplished by subtracting global minimum fluorescence from samples and determining the point in the cycle which amplification reached exponential amplification phase. Default settings were accepted for each program to ensure reproducibility. The BioRad CFX Manager used default settings of single threshold for $Cq$ determination and baseline subtracted curved fit for each run. The Opticon Manager used default settings of subtract baseline via global minimum, which estimated the threshold as being between 0.019 and 0.028. Gene expression values were determined as normalized mRNA levels using the following equation ($\Delta Ct$): $2^{-\Delta Ct}$; where $\Delta Ct$ is: (target $Ct$—actin $Ct$) (*Schmittgen & Livak, 2008*). Actin expression levels were determined to be consistent across all samples and served as an internal amplification control to use for expression normalization. Data from $\Delta Ct$ did not exhibit normal distributions, so were log transformed (log $\Delta Ct$), to establish normal data distributions for statistical analysis. Two-way analysis of variance (ANOVA) followed by Tukey's Honestly Significant Difference post hoc test (*base, R Core Team, 2013*) were performed on log $\Delta Ct$ for each target ($p < 0.05$).

# RESULTS

## Gene expression analysis

Without considering separate populations, acute heat shock resulted in statistically significant increases in expression of coactivator-associated arginine methyltransferase 1 (CARM1) ($n = 24$ oysters per treatment, ANOVA, $df = 2$, Tukey's HSD $p = 0.00007$) (Fig. 1) and Histone 2AV (H2AV) ($n = 24$ oysters per treatment, ANOVA, $df = 2$, Tukey's HSD $p = 0.001$) (Fig. 2). A statistically significant increase in expression of tumor necrosis factor receptor-associated factor 3 (TRAF3) ($n = 24$ oysters per treatment, ANOVA, $df = 2$, Tukey's HSD $p = 0.008$) (Fig. 3) occurred upon exposure to mechanical stress.

There was a clear difference in response to mechanical stress in oysters from Oyster Bay as compared to oysters from Dabob and Fidalgo Bays. Specifically, upon heat shock, H2AV

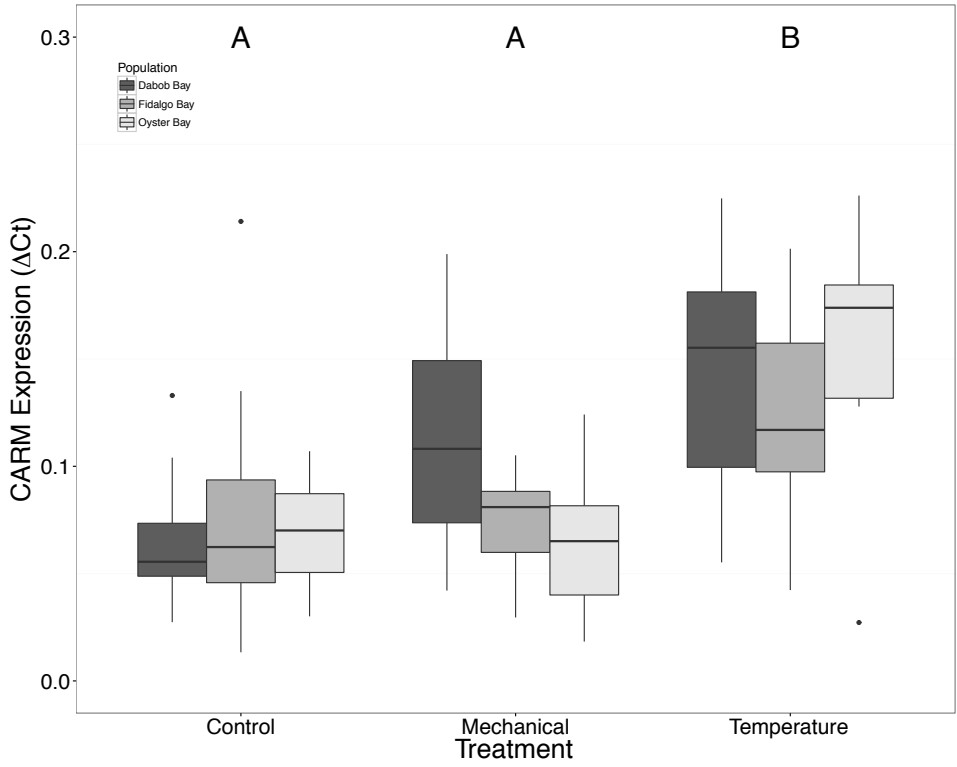

**Figure 1** **Expression of CARM1 mRNA.** Median $\Delta Ct$ indicated by line in middle of box plot. Shaded boxes are 2nd and 3rd quartile groups. Lines are 1st and 3rd quartiles. Dots indicate outside values. Capital letters indicate significant differences ($p < 0.05$) between overall treatment groups ($n = 24$ animals per treatment). No statistical differences ($p > 0.05$) were observed between populations ($n = 24$ animals per population), nor within a given population ($n = 8$ animals per treatment).

expression in oysters from Oyster Bay increased ($n = 8$ oysters per population, ANOVA, $df = 4$, Tukey's HSD = 0.05) (Fig. 2) when compared to the control. When exposed to mechanical stress, bone morphogenic protein 2 (BMP2) ($n = 8$ oysters per population, ANOVA, $df = 4$, Tukey's HSD $p = 0.03$) (Fig. 4) and growth-factor receptor bound protein 2 (GRB2) ($n = 8$ oysters per population, ANOVA, $df = 4$, Tukey's HSD $p = 0.03$) (Fig. 5) expression was decreased in the Oyster Bay population, whereas there was no significant differences in responses in the other populations. Additionally, significant interactions were identified between population and treatment in both BMP2 and GRB2 ($p < 0.05$).

There was no statistical difference in expression in Peptidoglycan recognition protein 1 (PGRP), toll-like receptor 2 type 1 (TLR), and prostaglandin E2 receptor EP4 subtype (PGEEP4) (Figs. 6–8, respectively) within any comparison. Heat shock protein 70 gene expression was significantly different between temperature and mechanical stress ($n = 24$ oysters per treatment, ANOVA, $df = 4$, Tukey's HSD $p = 0.006$) (Fig. 9).

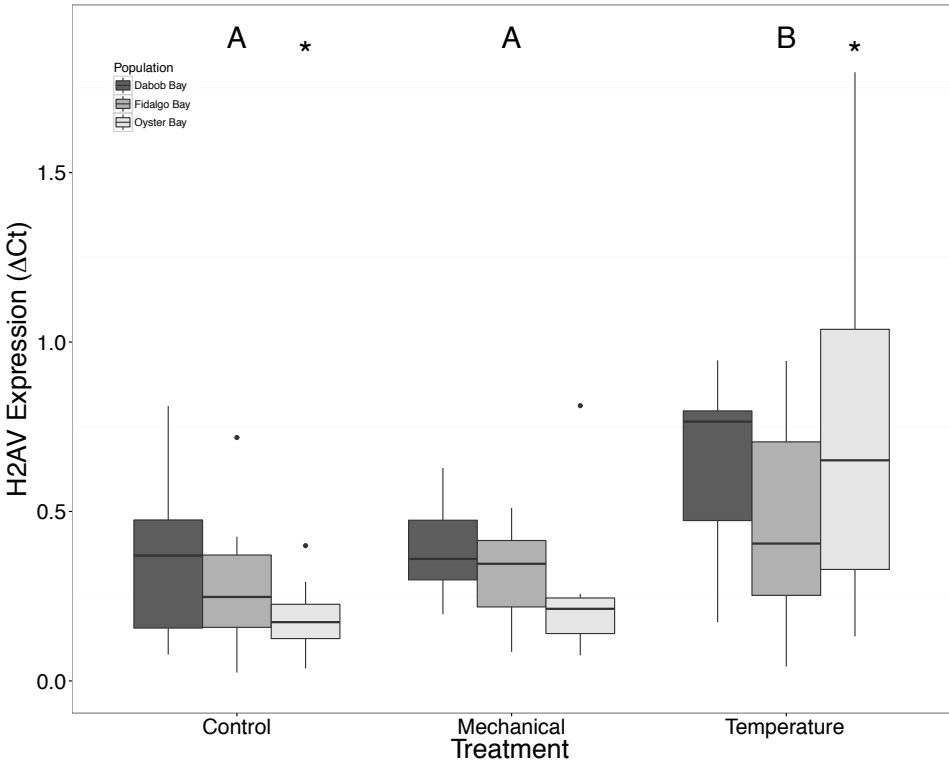

**Figure 2** **Expression of H2AV mRNA.** Median $\Delta Ct$ indicated by line in middle of box plot. Shaded boxes are 2nd and 3rd quartile groups. Lines are 1st and 3rd quartiles. Dots indicate outside values. Asterisks indicate significant differences ($p < 0.05$) between treatments within a population ($n = 8$ animals per treatment). Capital letters indicate significant differences ($p < 0.05$) between overall treatment groups ($n = 24$ animals per treatment). No statistical differences ($p > 0.05$) were observed between populations ($n = 24$ animals per population).

## DISCUSSION

### Response to temperature stress

The response of *Ostrea lurida* to acute heat stress appears to include an alteration in gene regulatory activity and the innate immune response, as indicated by significant increases of H2AV (Fig. 2) and CARM1 (Fig. 2) gene expression one hour post-temperature stress.

Histone 2AV, H2AV, is a variant of the histone H2A protein. This variant has been shown to act as a transcription promoter agent as well as assist with heterochromatin formation. *Truebano et al. (2010)* characterized changes in transcription in Antarcticclams, *Laternula elliptica*, and found that an H2A variant was significantly upregulated under heat stress conditions (3 °C for 12 h). In addition to involvement in the heat stress response, histone H2A has been shown to exhibit antimicrobial properties in three invertebrates: two marine invertebrates (Pacific white shrimp and scallops; *Patat et al., 2004*; *Li et al., 2007a*), as well as in a freshwater shrimp (*Arockiaraj et al., 2013*). In *D. melanogaster,* H2Av is phosphorylated in response to DNA damage (*Madigan, Chotkowski & Glaser, 2002*) to inhibit apoptosis, suggesting an additional role in in cellular survival.

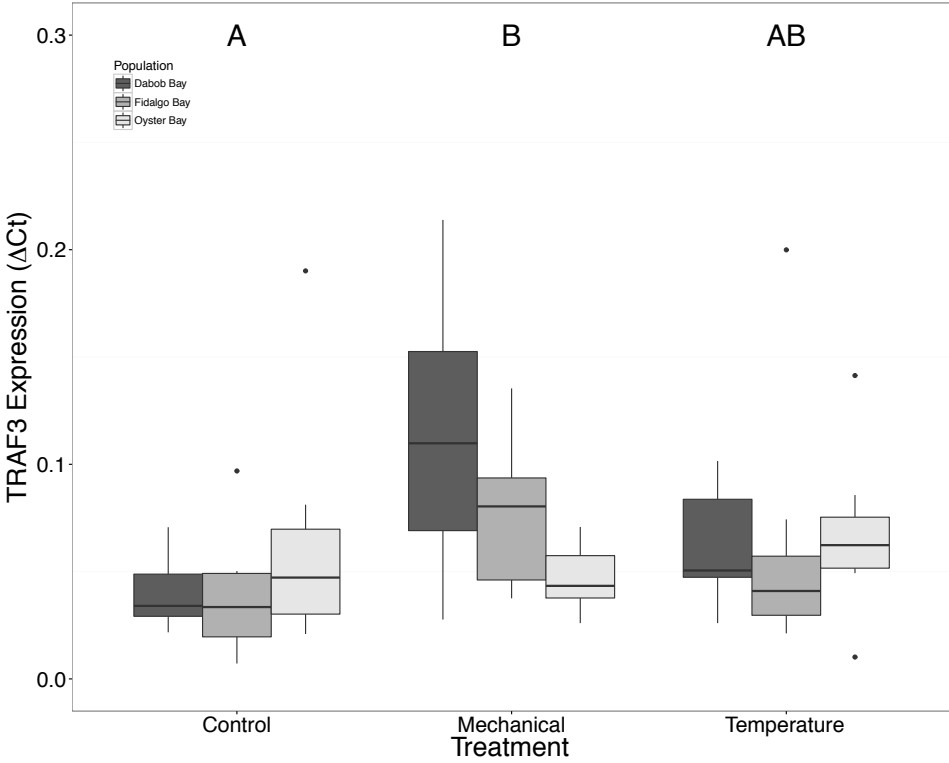

**Figure 3 Expression of TRAF3 mRNA.** Median $\Delta Ct$ indicated by line in middle of box plot. Shaded boxes are 2nd and 3rd quartile groups. Lines are 1st and 3rd quartiles. Dots indicate outside values. Capital letters indicate significant differences ($p < 0.05$) between overall treatment groups ($n = 24$ animals per treatment). No statistical differences ($p > 0.05$) were observed between populations ($n = 24$ animals per population), nor within a given population ($n = 8$ animals per treatment).

Coactivator-associated arginine methyltransferase 1, CARM1, is involved in transcriptional activation via methylation of histones (*Chen et al., 1999*; *Lee et al., 2005*). This in turn affects the ability of transcription factors to bind and transcription to proceed. It is possible that increases in CARM1 expression could indicate that overall gene regulatory activity is increased in response to temperature stress. Our results are similar to those of *Wang et al. (2011)* where researchers described an increase in expression of Histone-arginine methyltransferase in the sea cucumber, *Apostichus japonicus*, after experiencing 25 °C temperatures for seven days. The authors suggested that this was due to an induced dormancy and lower metabolic rate to provide resources for stress resilience. CARM1 is also a component of the cellular immune response, as it has been identified as a regulator of NF-kB (*Covic et al., 2005*). Thus another explanation is that acute heat could possible impact the immune response, likely in a negative manner. Future work, that would be relevant to restoration activities, should increase the number of stressors examined in oyster to include pathogens.

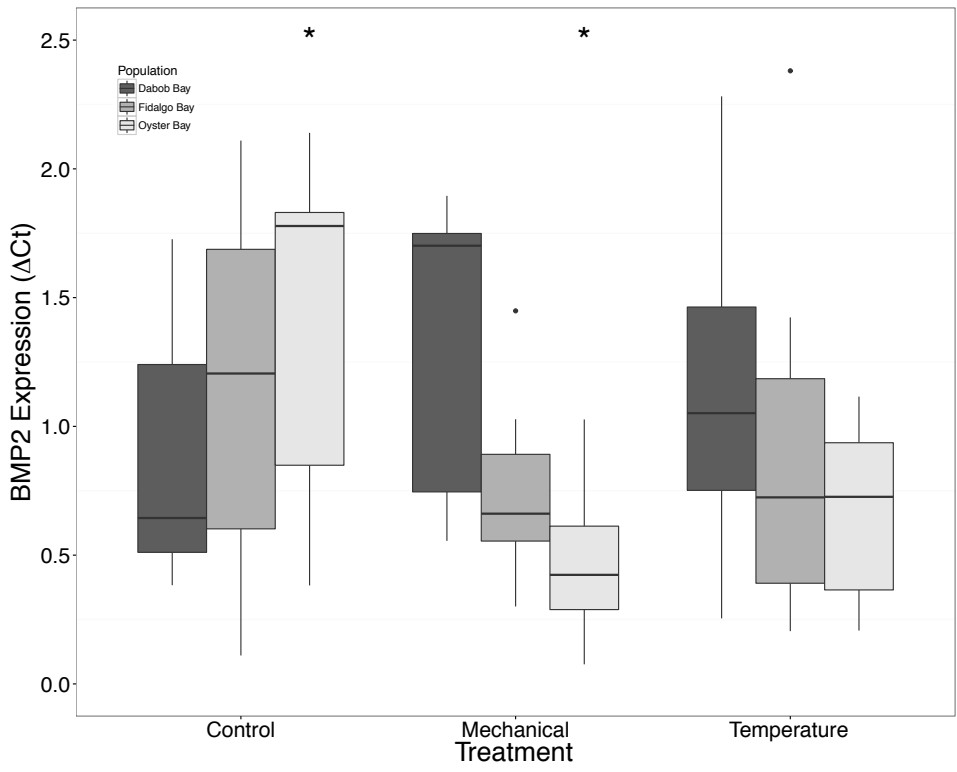

**Figure 4** **Expression of BMP2 mRNA.** Median $\Delta Ct$ indicated by line in middle of box plot. Shaded boxes are 2nd and 3rd quartile groups. Lines are 1st and 3rd quartiles. Dots indicate outside values. Asterisks indicate significant differences ($p < 0.05$) between treatments within a population ($n = 8$ animals per treatment). No statistical differences ($p > 0.05$) were observed between populations ($n = 24$ animals per population), nor between treatments ($n = 24$ animals per treatment).

Increases in HSPs are often observed in response to stress, but this study only found a significant difference of mRNA expression of HSP70 in the Oyster Bay population between mechanical and heat stresses (Fig. 9). *Brown et al. (2004)* found the maximum HSP expression in *O. lurida* occurred 24–48 h post exposure to 39 °C. The absence of a strong response of HSP70, relative to the control group, could be related to temporal changes in expression or an isoform-specific response, as there are many genes in this gene family, particularly in oysters (*Clegg et al., 1998*; *Piano et al., 2005*). Mediterranean mussels, *Mytilus galloprovincialis*, have shown different isoforms of heat shock proteins and cognates that have differential expression patterns caused by heat, mercury exposure, and chromium exposures stressors suggesting that the isoforms have slightly different functions (*Franzellitti & Fabbri, 2005*). Additionally, there are members of the HSP70 gene family that are constitutively expressed and do not exhibit increases in mRNA in response to heat stress (*Sorger & Pelham, 1987*; *Somji et al., 1999*). Without a sequenced genome for *Ostrea lurida*, combined with utilizing an incomplete transcriptome, it is difficult to ascertain how many isoforms might exist, as well as the number of alternatively spliced products. Upon

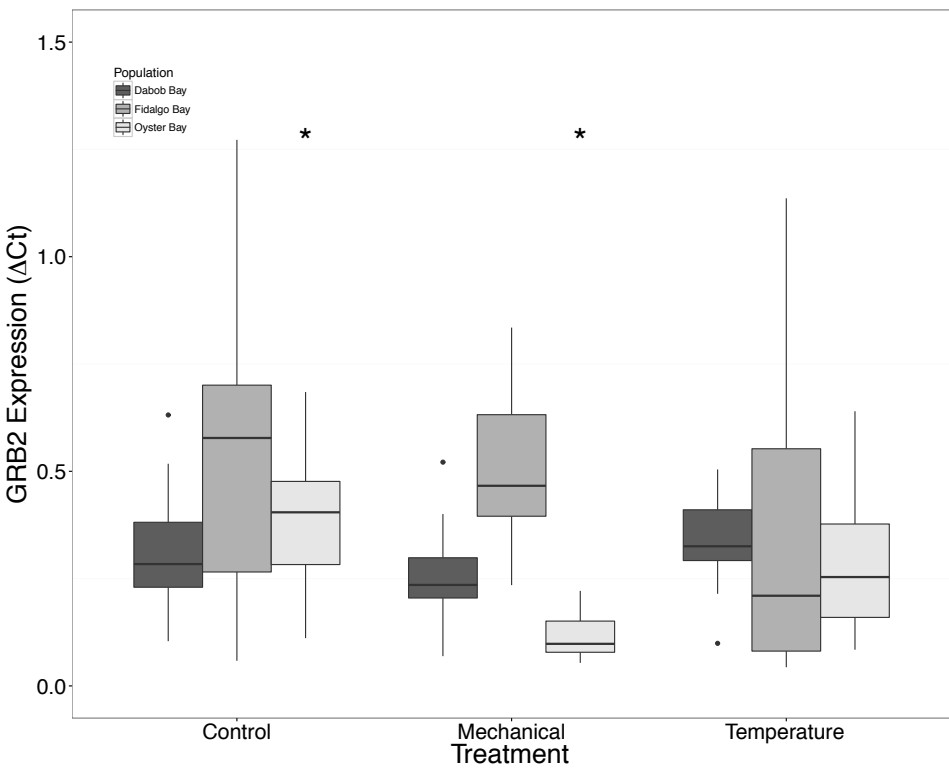

**Figure 5 Expression of GRB2 mRNA.** Median $\Delta Ct$ indicated by line in middle of box plot. Shaded boxes are 2nd and 3rd quartile groups. Lines are 1st and 3rd quartiles. Dots indicate outside values. Asterisks indicate significant differences ($p < 0.05$) between treatments within a population ($n = 8$ animals per treatment). No statistical differences ($p > 0.05$) were observed between populations ($n = 24$ animals per population), nor between treatments ($n = 24$ animals per treatment).

addition of new genomic resources the entire family of molecular chaperones could be examined and compared across populations.

## Response to mechanical stress

Mechanical stress increased expression of inflammation-related target genes. In all populations, there was a significant increase in immune system-related responses seen via the expression of tumor necrosis factor receptor-associated factor 3, TRAF3 (Fig. 3), which is involved in internal tissue damage recognition and apoptosis. The main function of TRAF3 is to assist in cell death initiation caused by stress conditions within tissues (*Arch, Gedrich & Thompson, 1998*). Upregulation in relation to mechanical stress could be akin to inflammation occurring due to edema from the mechanical stress and used to remove damaged cells as suggested by *Roberts, Sunila & Wikfors (2012)* when *C. virginica* were exposed to mechanical stress. Significant differences in expression of other immune system targets such as PGRP, TLR, and PGEEP4 were not detected (Figs. 6–8, respectively), but other studies have found that the time scale for expression may vary (*Meistertzheim et al., 2007*; *Farcy et al., 2009*).

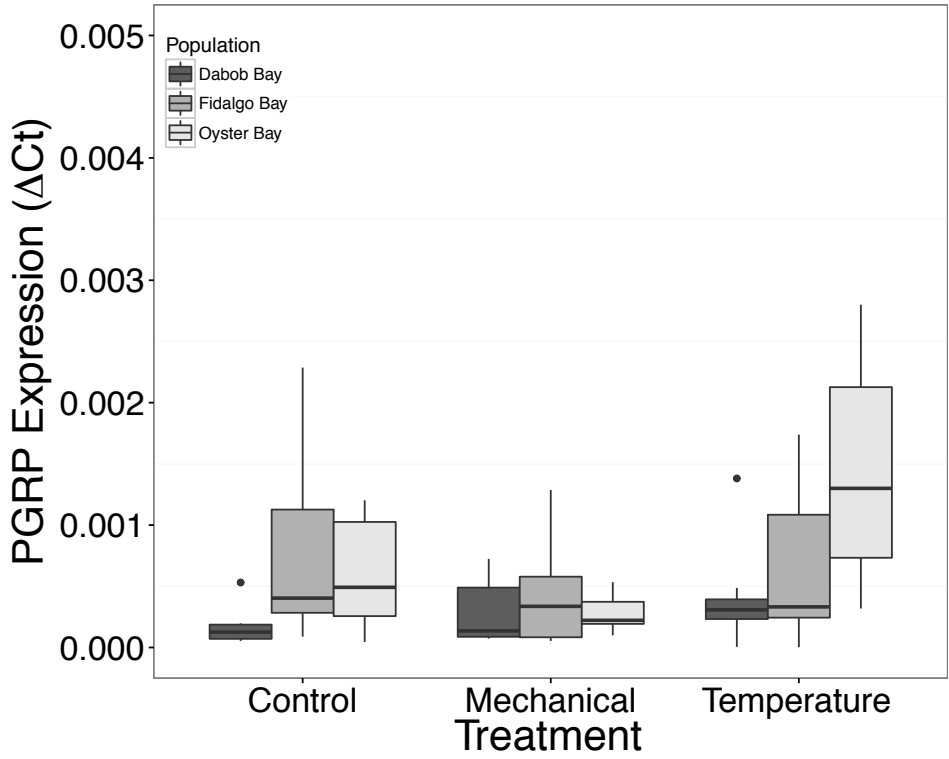

**Figure 6** **Expression of PGRP mRNA.** No statistical difference observed between treatments, nor between populations. Median $\Delta Ct$ indicated by line in middle of box plot. Shaded boxes are 2nd and 3rd quartile groups. Lines are 1st and 3rd quartiles. Dots indicate outside values. No statistical differences ($p > 0.05$) were observed within populations between treatments ($n = 8$ animals per treatment), between populations ($n = 24$ animals per population), or between treatments ($n = 24$ animals per treatment).

## Population differences

We suspected that the Dabob Bay population would have demonstrated a more pronounced response to stress as this population is subjected to greater environmental fluctuations with respect to salinity and temperature (*Heare et al., 2017*). Contrary to our hypothesis, oysters from Oyster Bay were the only population that exhibited a difference in gene expression in response to mechanical or heat stress. Oysters from Oyster Bay parents showed an increase in H2AV expression during heat stress as compared to control (Fig. 2), a decrease in BMP2 and GRB2 upon mechanical stress (Figs. 4 and 5, respectively), and differences in HSP70 expression between heat and mechanical stresses (Fig. 9). Given the putative function of H2AV in transcriptional regulation (Table 1), the increase in expression could be indicative of the role of this protein in controlling the molecular response to stress. Bone morphogenic protein 2, BMP2, and growth-factor receptor bound protein 2, GRB2, were significantly decreased in expression which could be indicative of growth inhibition. Both genes are related to growth and development of tissues, with BMP2 being a pre-cursor to osteoblastic cells that produce shell (*Pereira Mouriès et al., 2002*) and GRB2
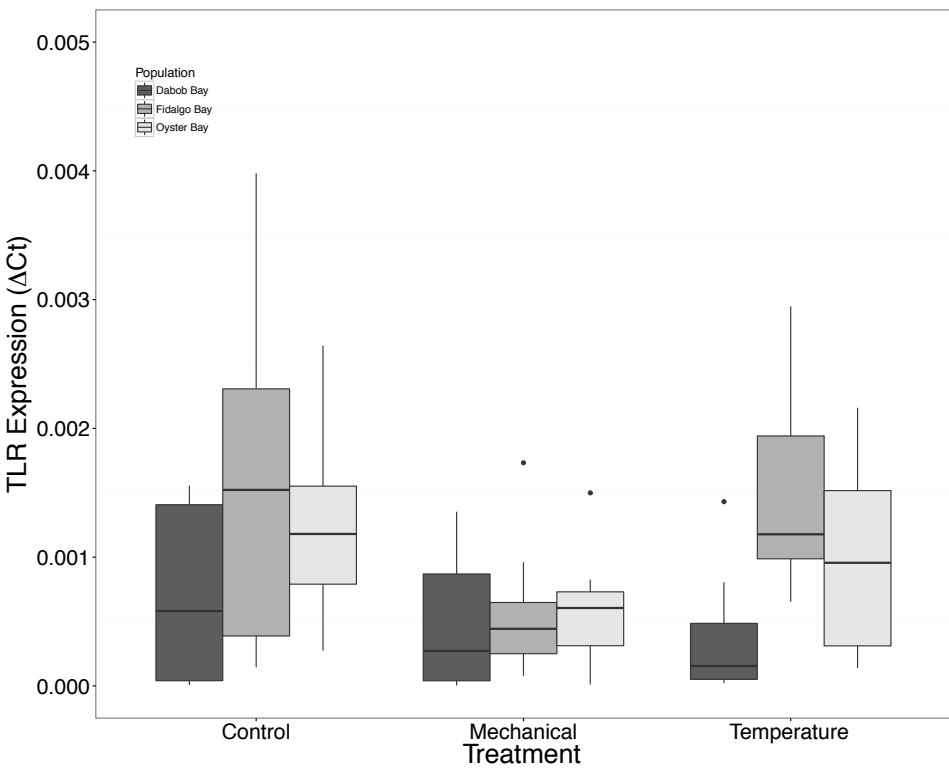

**Figure 7** **Expression of TLR mRNA.** No statistical difference observed between treatments, nor between populations. Median $\Delta Ct$ indicated by line in middle of box plot. Shaded boxes are 2nd and 3rd quartile groups. Lines are 1st and 3rd quartiles. Dots indicate outside values. No statistical differences ($p > 0.05$) were observed within populations between treatments ($n = 8$ animals per treatment), between populations ($n = 24$ animals per population), or between treatments ($n = 24$ animals per treatment).

is used for signal transduction between cells during growth phases (*Oda et al., 2005*). By down-regulating these targets, this may be an effort to reduce energetically costly processes in favor of processes that promote survival during stress events. Organisms faced with stress are often required to reallocate energy resources to homeostasis-related functions in an effort to improve long-term survival of the species (*Sokolova et al., 2012*). This change in expression coupled with the up-regulation of H2AV (Fig. 2) is in accord with the idea of shifting priorities for stress resilience.

Interactions were identified between population and treatment for both BMP2 and GRB2. Differences between gene expression in control and mechanical stress in the Oyster Bay population are driving this interaction for both genes. Although statistical interactions of this nature are difficult to interpret, it could be related to fact the Oyster Bay population is from a relatively "low-stress" environment (i.e., abundant food and less-pronounced temperature fluctuations).

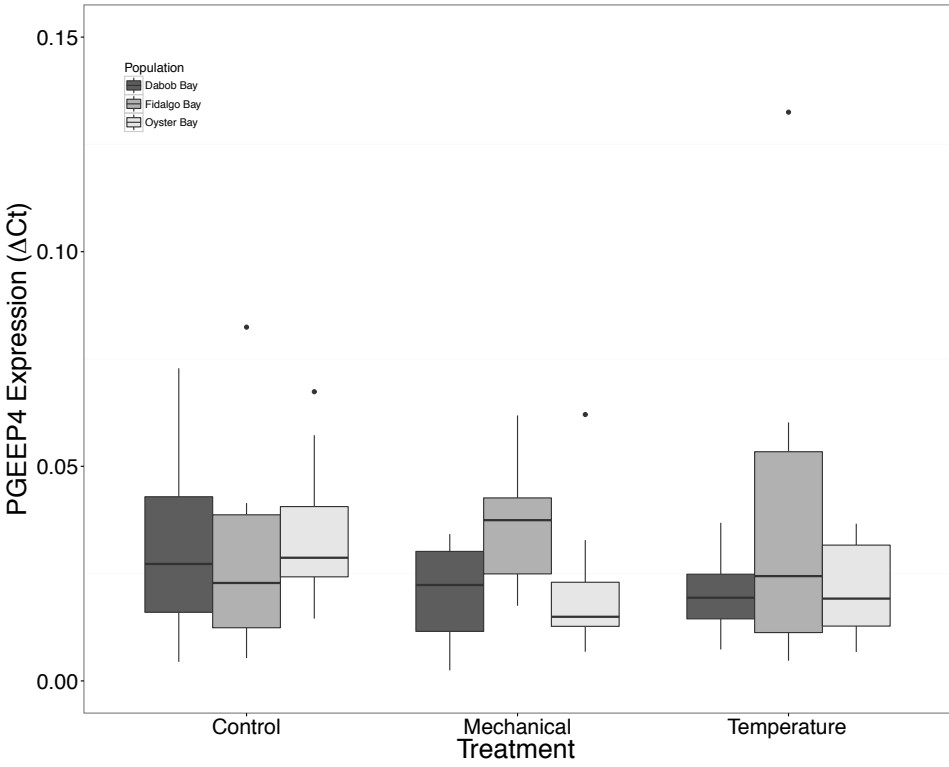

**Figure 8 Expression of PGEEP4 mRNA.** No statistical difference observed between treatments, nor between populations. Median $\Delta Ct$ indicated by line in middle of box plot. Shaded boxes are 2nd and 3rd quartile groups. Lines are 1st and 3rd quartiles. Dots indicate outside values. No statistical differences ($p > 0.05$) were observed within populations between treatments ($n = 8$ animals per treatment), between populations ($n = 24$ animals per population), or between treatments ($n = 24$ animals per treatment).

## CONCLUSIONS

The gene expression pattern differences observed here with oysters from Oyster Bay coupled with corresponding field-based observation that this population has the greatest reproductive activity (*Heare et al., 2017*), could indicate this population has a greater ability to effectively respond to stress. Another way to consider this is that the Oyster Bay population has a relatively higher degree of phenotypic plasticity, or more specifically, an elevated rate of phenotypic change (*Angilletta et al., 2003*). The gene expression data indicates a clear population-level stress response, and lack of differential response in other populations that suggests shifts in energy balance. Some possible explanations for this relatively rapid response include a more sensitive cell-signaling system (ie cytokines) or a more robust transcription initiation process. *Yao & Somero (2012)* observed higher heat stress tolerance in *M. galloprovincialis* than *M. californius* likely due to their ability to maintain cell signaling through the production of phosphor-p38-MAPK kinases, which may be how the Oyster Bay population is able to quickly respond to stress. This ability to quickly respond to stress may be due to increased fitness in Oyster Bay, however more research is needed to identify the link between gene expression and performance. Based on

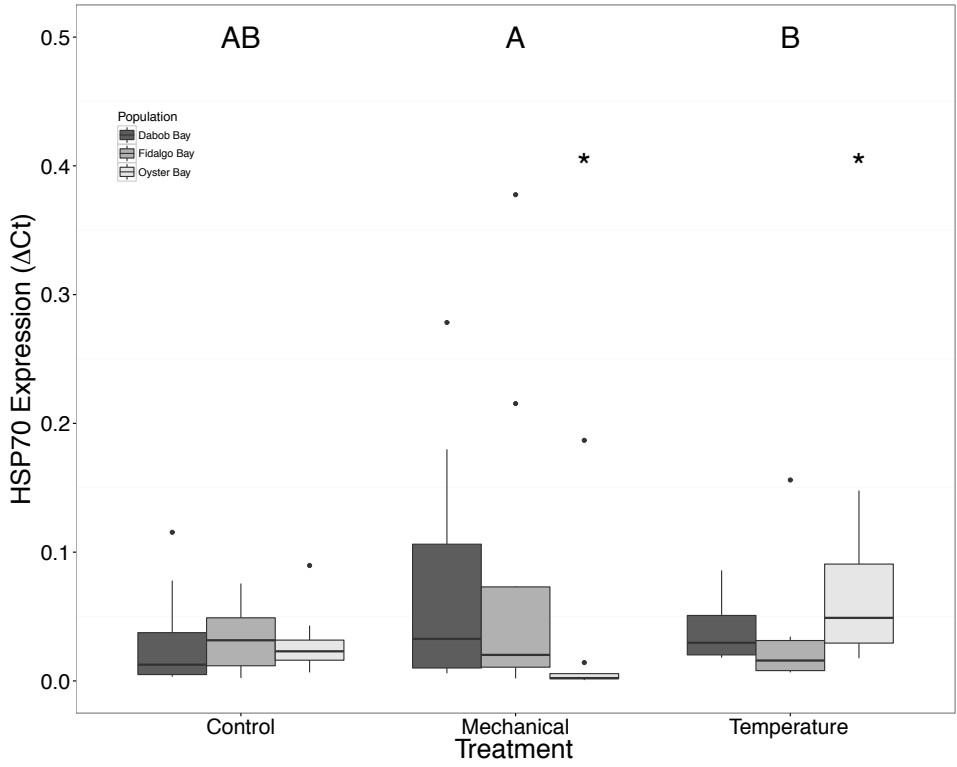

**Figure 9** **Expression of HSP70 mRNA.** Median $\Delta Ct$ indicated by line in middle of box plot. Shaded boxes are 2nd and 3rd quartile groups. Lines are 1st and 3rd quartiles. Dots indicate outside values. Asterisks indicate significant differences ($p < 0.05$) between treatments within a population ($n = 8$ animals per population). Capital letters indicate significant differences ($p < 0.05$) between overall treatment groups ($n = 24$ animals per treatment). No statistical differences ($p > 0.05$) were observed between populations ($n = 24$ animals per population).

earlier field work, this could be directly linked to increased larval production, and processes allocating limited resources into reproduction (*Heare et al., 2017*). This trait could certainly be perceived as advantageous for restoration purposes. Caution should be used in using non-local stocks when structure exists, as it is possible to have supplemented oysters out-compete the native population or to create hybrids that are ultimately less fit than the native counter parts (*Camara & Vadopalas, 2009*). Both such phenomena decrease overall genetic diversity leaving the remaining population to be less robust for future challenges and possibly leading to local extirpation.

Another interpretation of gene expression patterns in the Oyster Bay population is that the differences observed upon stress exposure are not indicative of an effective response that has been selected for, but rather indicative of plasticity. In other words, the change in gene expression upon stress is representative of a phenotype that is tolerable to a wide range of pressure. At one level the ability to achieve a number of phenotypes with a given genotype could be advantageous, particularly in a rapidly the changing environment. There is a paradox in the fact that too much plasticity negates the ability of natural selection to function. Populations with high phenotypic plasticity become deprived of

negative selection and thus are often able to survive in rapidly changing environments as long as the changes are consistent and somewhat predictable. However, with this increased adaptive ability, genetic diversity and adaptation become limited within a population that may be unable to properly respond to novel challenges in the future (*Crispo, 2008*). Alternatively, the Baldwin effect may enhance long-term genetic diversity by allowing species to colonize novel habitats and, with phenotypic plasticity, eventually genetically diverge from the source population through induced genetic adaptations (*Crispo, 2007*). For longterm restoration of *O. lurida* populations in Puget Sound, understanding the genetic differences and phenotypic plasticity of individual populations will help determine proper supplementation procedures for existing and historic habitats.

## ACKNOWLEDGEMENTS

The authors would like to thank an anonymous reviewer and Marta Gomez-Chiarri for their helpful insight and feedback upon initial submission of this manuscript for publication. We would also like to thank Puget Sound Restoration Fund for providing the oysters used in these experiments.

### Funding

This work was funded by Washington Sea Grant, University of Washington, pursuant to National Oceanographic and Atmospheric Administration Award No. NA10OAR4170057 Projects R/LME/N-3. The funders had no role in study design, data collection and analysis, decision to publish, or preparation of the manuscript.

### Grant Disclosures

The following grant information was disclosed by the authors:
National Oceanographic and Atmospheric Administration: NA10OAR4170057 Projects R/LME/N-3.

### Competing Interests

The authors declare there are no competing interests.

### Author Contributions

- J. Emerson Heare conceived and designed the experiments, performed the experiments, analyzed the data, contributed reagents/materials/analysis tools, wrote the paper, prepared figures and/or tables, reviewed drafts of the paper.
- Samuel J. White performed the experiments, wrote the paper, prepared figures and/or tables, reviewed drafts of the paper, processed samples.
- Brent Vadopalas conceived and designed the experiments, analyzed the data, statistics consultations.
- Steven B. Roberts conceived and designed the experiments, analyzed the data, contributed reagents/materials/analysis tools, wrote the paper, prepared figures and/or tables, reviewed drafts of the paper.

## Data Availability

Github: https://github.com/RobertsLab/paper-Olurida-gene.

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
