# Peer review of "Differential response to stress in Ostrea lurida as measured by gene expression"

_PeerJ, doi:10.7717/peerj.4261_

## Round 0.1 · original submission · Major Revisions

Please, consider the suggestions from the reviewers in the new version of your manuscript.

Reviewer 1 ·

Basic reporting

BASIC REPORTING

Major considerations

The manuscript is generally well-written and the language clear and generally unambiguous (specific problems are addressed further in this review.) There are some minor typos and occasional grammatical errors that can easily be caught in revision. The structure of the manuscript conforms to the PeerJ standard.
The Intro & background give the reader context and are well-referenced. However at times this section seems to delve in to what may be too much detail given the rather general scope of PeerJ.

The hypothesis is well-constructed and supported based on the background information, but commentary on reproductive cycle specifics may not be required based on the exact gene expression data collected and the discussion later in the paper.

The authors set out to test the hypothesis that one geographically separate population may have a more pronounced stress response than the others and mention relation to the Baldwin effect and reproductive performance in the discussion. There is mention of local efforts to restore O. lurida and reproductive performance is obviously a consideration for such efforts. For the generalist reader more explicitly mentioning these models and their relevance to our understanding of biology in general as well as their relation and application to restoration efforts may help the reader better understand the intentions and relevance of the research and the rationale for the general research design. Indeed, the rationale is in the manuscript, and understandable to the reader, but scattered and not emphasized in an obvious way. Currently the authors are under-selling the value of their work and relevance. A bit of reorganization and focus on greater biological questions at hand or potential application of the knowledge would remedy this.

The figures are all relevant and generally well-presented. The few concerns I do have are mentioned below.

Minor considerations

It would help if the reader is referred to Figure 1 in the introduction describing the different populations rather than in the Material and Methods section.

Experimental design

EXPERIMENTAL DESIGN

The specific experimental design directly tests the hypothesis that oysters from Dabob Bay will demonstrate a more pronounced response to stress via changes in gene expression compared to oysters from other geographic locations. This is done through comparison of mRNA levels of a battery of different O. lurida genes in animals taken from three different geographic locations that were placed under temperature or mechanical stress. The mRNAs studied were predicted to be involved in stress response based on sequence similarity to genes with well described functions in other systems.

The methods with respect to the biochemistry are described in sufficient detail for replication by investigators experienced with RT-qPCR although minor clarifications are required.

The data analysis, however, would benefit if the authors were more explicit with how changes in gene expression were calculated and why. This is especially true with respect to what a sample (n=1) represents in the data sets. It would be appreciated if the authors would clarify the following:

Major considerations

Line 117-127: Would the authors please clarify if tissue was pooled from animals. Explicit descriptions of what one data point represents would assist with interpretation of data and assessment of its robustness. Further, were technical replicates performed, or was only one RT-qPCR reaction performed per unique animal per gene?

Clearly “stating n= X animals per condition” for example in the results section or in the figure legend would make this crystal clear to the reader.

Line 174: This table is useful, but please comment on how gene function was inferred from sequence. Is this from GO terms based on closest known homolog? Is this manual or semi-automated curation? Additionally, are there potential paralogues for these genes and why was one picked over another? The authors even suggest that the HSP70 transcript they chose may not actually be the true HSP70-like protein in other organisms given the lack of transcript change in response to heat stress.
In the last paragraph of the introduction the authors state “A suite of genes was selected based on their predicted function (gene regulation, immune response, and growth).” Binning the genes in to these categories in the table helps connect the experimental design to the hypothesis and may make discussion of the results easier.

Line 187-201: Even though Real Time Quantitative PCR as a commercially available tool has been around nearly longer than some of the youngest current grad students have been alive, data analysis and reporting is still inconsistent between groups and multiple methods for absolute quantitation based on normal curves or relative levels to a reference gene are in use. Statistical analysis of these data represents even more of a “Wild West” situation.

The authors reference the Schmittgen and Livak, 2008 paper. Schmittgen and Livak offers a number of different analysis methods. The fold change=2−ΔΔCt method where 2−ΔΔCt = [(Ctgene of interest - Ctactin)treatment - (Ctgene of interest - Ctactin)control] is often used by other groups and could also be appropriate for this type of data as best I can tell. Please explicitly state why the normalized mRNA levels are calculated as ΔCt= (target Ct – actin Ct) as well as why a logΔCt were used for statistical analysis. In the results section the y-axis is labeled as the data being plotted at a ΔCt value. The statistical calculations being performed on logΔCt value but the graphs showing just the ΔCt can be a bit confusing. It’s not so much that there’s an inherent problem with how the data were analyzed, but rather just being clear why it was done this way. Overall, the treatment of the data as ΔCt values, given the way the experiments were designed and run, is still better than doing fold-change calculations and statistics with the fold-change without efficiency correction and linearity controls as many do. Kudos to the authors for avoiding this pitfall.

Line 204-207: These data are mentioned in the discussion but it’s unclear from the introduction as to why these experiments were done in the first place. What direct relevance does this have on the rest of the data and or with respect to how this work falls within the greater body of the literature?

Minor considerations

Line 126-127: Was the tissue simply stored in RNAzol or was it lysed in RNAzol and one of the intermediary steps used for storage at -80oC? The manufacturer’s protocol has a number of convenient stop points but storage of whole tissue in RNAzol is not in any protocols I’ve seen or performed myself.

Line 148-149: Please comment on which primers were used to detect genomic DNA via PCR. Were primers targeted to intronic or mRNA non-coding regions? Or were primers generating an amplicon spanning an exon junction used?

Line 152-157. Awkward phrasing.

Line 161: Reference renders and links strangely in PDF. This may be something the editor needs to deal with.

Lines 159-168: Please comment on whether the priming sites for the primers were known to contain SNPs that may have affected hybridization efficiency. This should be available since the Authors’ lab appears to have made its own transcriptome and have access to raw sequencing reads. Since the authors are not using highly inbred populations and these populations are originating from different geographical locations where some degree of genetic drift, founder effects, selective pressures, etc. may have generated SNPs between groups there may be variability in the mRNA sequences. A single SNP in a primer binding sequence is enough to dramatically affect replication efficiency and the downstream relative abundance calculations. Even within highly inbred lab strains of fruit flies or C. elegans SNPs are observed within a lab’s own stocks and between labs- even in protein coding sequence or regions of high evolutionary conservation!

Validity of the findings

VALIDITY OF THE FINDINGS

Specific comments on the Results section

If n=8 animals as the methods suggest then the data appear to be generally robust, statistically sound, and controlled. No concerns there except for those raised above in terms of data representation and explicit statement of what n=1 constitutes. Interpretation of two-way ANOVAs can be tricky, but I think the authors have done a generally good job here. Lack of significant difference is also noted and is relevant to the hypothesis and the discussion of the results.

What may make the results section easier to interpret for a more generalist reader is if full gene names were given as they are in the discussion. Possibly the table currently in the methods section could be split up in to one table just for primer design and another put in to results for gene name and inferred function from sequence similarity or existing literature.

Specific comments on the Discussion section

In general, connecting changes in mRNA level of the battery of genes studied with existing literature is well done. The inferences are sane and based in literature findings without speculation that raises any flags. However, many of these genes have been very well-studied in model genetic organisms like mouse, Drosophila or C. elegans. It may be worth citing some of that literature as well. In some places the authors already do this. Are there potentially more overarching biological phenomena you’re observing here that are worth noting?

Given some of the literature cited in the discussion, I get the feeling that some of the mRNA targets were chosen based on what was already known in other marine organisms. That’s an understandable way to pick targets. If that’s true, maybe this should be brought up in the methods or results section when discussing what targets were chosen and why. This would also help with proposing gene function from sequence similarity arguments.

Lines 226-237: Again, the discussion on mortality seems disconnected from the specific hypothesis of this study. More needs to be done to connect this to the rest of the paper.

Lines 258-267: The lack of change in the HSP70 orthologue is a bit worrisome. In many model systems the HSP70 transcriptional response should be in the timeframe of the experiments performed in this manuscript (minutes to hours.) There is some speculation as to isoform specific responses. From the transcriptome available, are different isoforms of this gene observed? Do the RT-qPCR primers target all or only a subset of them?

Specific comments on the Conclusion section

The authors conclude that the Oyster Bay population appears to have the greater ability to effectively respond to stress. This runs opposite to their initial hypothesis that oysters from Dabob Bay will demonstrate a more pronounced response to stress via changes in gene expression.

Lines 303-312: Conclusions are supported by the data - well done.

Lines 319-335: Maybe a little too speculative. The experiments done here don’t really directly test these very general models. Although, this discussion does put the results in a greater context. Depending on how the authors restructure the introduction some of this section could be put in the introduction.
Perhaps the style of writing and papers I’m used to reading in my field is different, but front-loading this in the introduction would make the general relevance and aims of this study much more clear. The line:
“For long term restoration of O. lurida populations in Puget Sound, understanding the phenotypic plasticity of individual populations will help determine proper supplementation procedures for existing and historic habitats,” succinctly sells me the rational for the research and experimental design.

Additional comments

Overall Reviewer’s Report

The hypothesis proposed is well-developed and the experiments to test the hypothesis are well designed and directly test the hypothesis. The authors’ discussion and conclusion are grounded in the data for the most part and made relevant with respect to the existing state of the field. Overall this work is of good quality on the raw science side of things. However, lack of clarity and consistency in some parts obfuscates the interpretation of the data as well as the general intentions and objectives of the work. These can be solved with revisions to the text of the manuscript. No additional experiments are required other than some low level bioinformatics with existing tools. In the following sections I have made recommendations where clarification or reorganization would fix these problems. I would happily re-review the manuscript upon revision.

·

Basic reporting

This research compares patterns of expression of selected genes in response to acute heat and mechanical stress in 3 populations of Olympia oysters. These populations were selected based on a previous study showing differences in performance in the field, as reported in a different manuscript (a preprint in PeerJ).

The research reported in this manuscript shows differences in gene expression in response to acute stress between populations, confirming that oysters from these locations show differences not only on performance, but also at the level of gene expression. These results have relevance for the restoration and aquaculture industries, indicating that more work should be done to evaluate the mechanisms behind those differences.

Experimental design

The experimental design is adequate, although the stressors used were very acute, resulting in 100% mortality of exposed oysters. The authors should indicate if those are consistent with stress happening in the field.

Validity of the findings

The research is well presented in general and the results are valid. I have attached a version of the manuscript with specific comments and suggestions. My major suggestion is to provide a more focused discussion that clearly differentiates between findings (differences in gene expression between populations in response to stress) and speculations regarding mechanisms behind those differences. The authors should also express that some caution should be taken into interpreting the relevance of these differences, since there was no difference between populations on their ability to survive these stressors (all oysters from all populations that were stressed died). Finally, I recommend that authors clearly indicate where the research should go next, and how that research will inform breeding in support of restoration efforts.

---

## Round 0.2 · accepted · Accept

Dear author,

Your manuscript has been accepted for publication. Thank you for submitting your work to this journal.

With kind regards,

·

Basic reporting

Pass: The authors have addressed all my comments to review satisfactorily. I consider that the manuscript it acceptable for publication.

Experimental design

Pass: The authors have addressed all my comments to review satisfactorily. I consider that the manuscript it acceptable for publication.

Validity of the findings

Pass: The authors have addressed all my comments to review satisfactorily. I consider that the manuscript it acceptable for publication.

Additional comments

The authors have addressed all my comments to review satisfactorily. I consider that the manuscript it acceptable for publication.